# AVSS: A NEW BENCHMARK FOR AIRPORT VIDEO SEMANTIC SEGMENTATION

## ABSTRACT

Airport video semantic segmentation is fundamental to airport surveillance applications, yet there currently lacks a specialized benchmark and algorithms for this task. In this paper, we introduce the first large-scale Airport Video Semantic Segmentation dataset (AVSS) for airport surveillance. AVSS comprises 18 common semantic categories at airports, and 250 videos, totaling over 140,000 frames with accurate manual annotations. AVSS covers a wide range of challenges for airport video surveillance, such as extreme multi-scale, intra-class diversity, inter-class similarity, etc. We analyze statistical information and evaluate 21 state-of-the-art (SOTA) semantic segmentation algorithms on AVSS. The significant performance degradation indicates that current models are far from practical application. Furthermore, we discuss how to develop video semantic segmentation algorithms for airport surveillance and the generalizability of AVSS to other tasks and datasets. AVSS serves as a research resource for airport semantic segmentation and a robustness evaluation tool for segmentation algorithms in practical applications. AVSS is available at www.agvs-caac.com/avss/avss.html.

## 1 INTRODUCTION

As global air passenger and cargo traffic rapidly increases, airports become more crowded, leading to frequent flight delays and occasional collisions. Thus, building intelligent airports is crucial to improving the efficiency and safety of airports. Intelligent airport applications, such as video docking and conflict warning systems, demand high-precision scene analysis to ensure safety and operational efficiency. These tasks rely on semantic segmentation to produce pixel-level masks, which are essential for accurately identifying and distinguishing different objects, such as cars, runways, and airplanes, within complex environments. Pixel-level segmentation provides precise boundaries, allowing for the detection of even slight overlaps or proximity between objects, which is crucial for preventing accidents (as shown in Figure 1). Bounding boxes alone cannot achieve this level of detail, as they only offer rough object localization and may miss critical spatial relationships and fine boundaries that could impact safety. So we observe the performance of existing semantic segmentation algorithms on airports.

Our experiments show that semantic segmentation algorithms performing well on public datasets suffer a significant performance drop in airport scenarios, with mean Intersection over Union (mIoU) decreasing by up to 78%. This decline is due to the unique semantic categories and structure of airports. For example, airplanes are challenging to segment due to their low-compact profile, unlike common categories such as buildings and vehicles. Additionally, airport imaging can reach kilometers, introducing extreme multi-scale challenges. These factors render public dataset-trained algorithms unsuitable for airport scenes, prompting us to develop a video semantic segmentation dataset specifically for airports and design specialized algorithms for this context.

This paper introduces a novel benchmark for semantic segmentation in airport video surveillance, called Airport Video Semantic Segmentation (AVSS). AVSS comprises 18 major semantic categories in airport ground (as shown in Figure 2), over 250 videos with more than 140,000 frames, addressing challenges like extreme multi-scale, intra-class diversity, inter-class similarity, etc. We manually annotated all frames to generate precise pixel-level ground truth. We evaluated 21 state-of-the-art segmentation models on AVSS, fine-tuning them for optimal performance. However, results

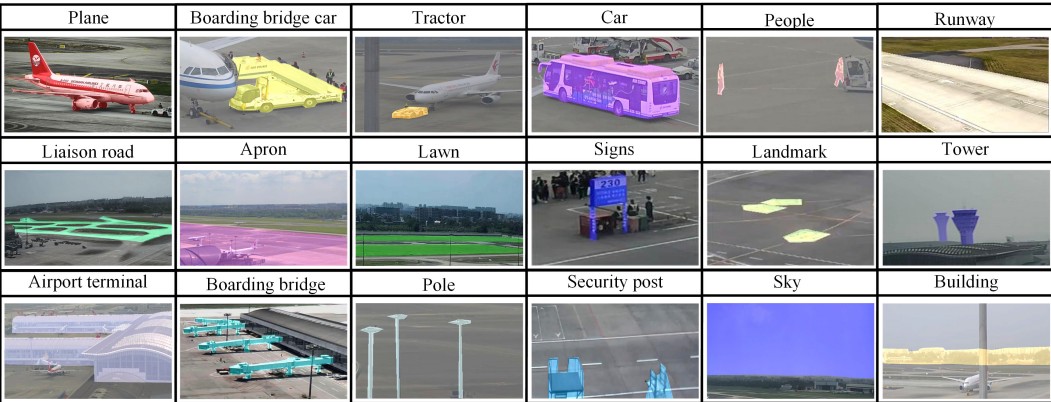

Figure 1: Conflict warnings application schematic.

Figure 2: 18 semantic classes in airports and corresponding ground truth.

show that current algorithms do not meet airport application demands. We also discuss future research directions for developing specialized algorithms for intelligent airport applications.

In conclusion, our primary contributions are as follows:

1. We establish the novel AVSS, providing a benchmark for airport semantic segmentation.

2. We evaluate the generalizability of AVSS and 18 SOTA segmentation algorithms on AVSS.

3. We propose principles for designing airport video semantic segmentation models.

## 2 RELATED WORK

We analyze the scale and type of existing semantic segmentation datasets and the performance and principles of existing semantic segmentation algorithms. The results illustrate the rarity and necessity of AVSS and semantic segmentation algorithms for airports. Next, We introduce semantic segmentation datasets and semantic segmentation algorithms in turn.

The semantic segmentation datasets are divided into image semantic segmentation datasets and video semantic segmentation datasets. Image semantic segmentation datasets focus on pixel-level classification of static images, including ADE20K Zhou et al. (2017), Pascal VOC Everingham et al. (2010), and COCO Lin et al. (2014). Video semantic segmentation datasets incorporate the temporal dimension, including Cityscapes Cordts et al. (2016), and VSPW Miao et al. (2021).

## 2.1 Semantic Segmentation Datasets

The semantic segmentation datasets are divided into image semantic segmentation datasets and video semantic segmentation datasets.

**Image semantic segmentation datasets** focus on pixel-level classification of static images, including ADE20K Zhou et al. (2017), Pascal VOC Everingham et al. (2010), and COCO Lin et al. (2014). ADE20K is for scene understanding, containing nearly 30,000 annotated images for semantic segmentation. Pascal VOC includes 20 semantic classes and one background class, comprising 11,000 images for recognition tasks. MS COCO is a large-scale dataset for object detection and segmentation, the images are from real-world scenarios, which adds diversity and realism. There are 80 object classes and one background class in COCO. **Video semantic segmentation datasets** incorporate the temporal dimension, including Cityscapes Cordts et al. (2016), and VSPW Miao et al. (2021). Cityscapes is the most widely used video semantic segmentation dataset of urban scenes, providing a wide variety of city street scene images with 5,000 high-quality pixel-level annotations. VSPW is the first large-scale multi-scene video semantic segmentation dataset. It is larger, more densely annotated, and covers more diverse scenes than other video segmentation datasets. It includes 3,536 videos and 251,632 semantic segmentation frames, including 124 semantic categories. However, current dataset types mainly focus on general scenes or urban streets, lacking datasets specifically for traffic scenes, especially airports.

## 2.2 Semantic Segmentation Algorithms

Current mainstream image semantic segmentation algorithms are designed for general scenes, but they perform poorly in specific scenarios, necessitating the development of semantic segmentation algorithms tailored to these specific contexts.

FCN Long et al. (2015) is the first CNN-based image semantic segmentation model, replacing fully connected layers with fully convolutional layers. PSPNet Zhao et al. (2017) introduced a pyramid pooling module to obtain multi-scale contextual information. DeepLab V3 Chen et al. (2017) further incorporated atrous spatial pyramid pooling and separable convolution to improve computational efficiency and segmentation accuracy. DeepLab V3+ Chen et al. (2018) optimized the decoder module for finer edge processing. ICNet Zhao et al. (2018) introduced a cascade branch network, gradually refining feature maps through cascading multiple branches, balancing segmentation speed and accuracy. HRNet Wang et al. (2020) proposed high-resolution sub-networks, presenting new solutions to resolution-limited and multi-scale problems. BiSeNetV1 Yu et al. (2018) and BiSeNetV2 Yu et al. (2021) utilize a dual-stream network to simultaneously capture spatial and semantic information, enabling real-time semantic segmentation. MobileNetV3 Howard et al. (2019) is a lightweight convolutional neural network designed for efficient performance on mobile and edge devices to enhance both speed and accuracy. Additionally, there are image semantic segmentation algorithms based on Transformers with global receptive field and better feature interaction, such as SETR Zheng et al. (2021), Swin Transformer Liu et al. (2021), Segmentor, and Segformer Xie et al. (2021). For video semantic segmentation, CFFM Sun et al. (2022a) and CFFM++ Sun et al. (2024) combine the dynamic and static contexts of the video to improve segmentation performance. And MRCFA Sun et al. (2022b) enables better temporal information aggregation by mining cross-frame affinities. Tube-Link Li et al. (2023) is a unified, near-online framework for video segmentation that enhances spatial-temporal tube tracking through attention and temporal contrastive learning. DVIS++ Zhang et al. (2023) proposes a decoupled framework for universal video segmentation, simplifying spatio-temporal modeling through separate segmentation, tracking, and refinement stages.

## 3 Benchmark

We present AVSS benchmark in terms of both data preparation and challenges in airports.

### 3.1 Data Preparation

Data preparation has three processes: dataset design, data collection, and data annotation.

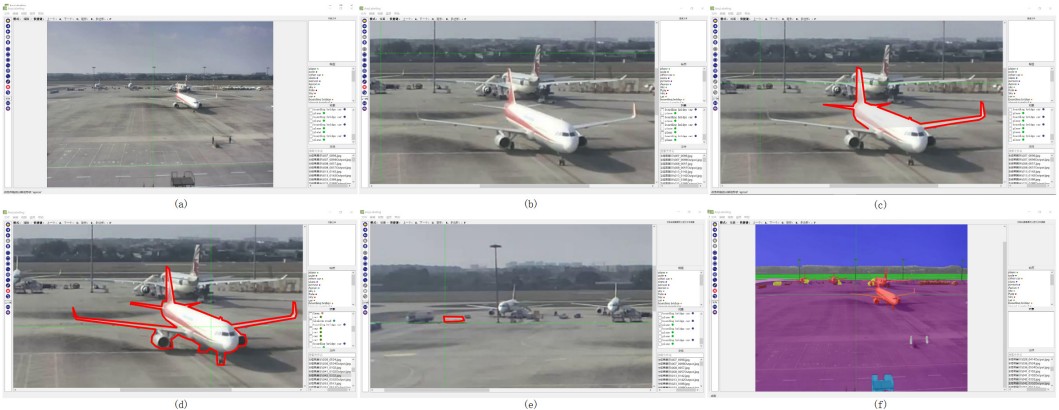

Figure 3: Data annotation process.

**Dataset Design**: The fundamental principle of dataset design is to cover as many semantic categories and variations as possible on the airport ground. Based on long-term observation and analysis, we identified 18 major semantic categories. These categories are divided into moving objects on the airport ground (airplanes, vehicles, and people), the airport ground itself (runways, liaison roads, aprons, lawns, landmarks, containers, and signs), inside the airport (tower, airport terminal, boarding bridge, pole, and security post), and external factors (sky, buildings, background).

**Data Collection**: Since airports are semi-militarized areas, we obtained data collection permissions through cooperation with the Civil Aviation Administration of China. The data collection sites are large international airports in Southwest China. To ensure data diversity, we recorded over 5,000 short videos within a three-month collection period, with an average duration of 15 seconds and a frame rate of 30 FPS. The data collection equipment included four fixed cameras and one PTZ camera, with resolutions varying in 1280×960, 1440×1080, and 1920×1080. During data collection, we increased data diversity by varying collection times and adjusting camera angles and focal lengths. We then strictly selected data based on diversity principles, picking 250 videos totaling about 140,000 frames to form the AVSS.

**Data Annotation**: The data annotation process is shown in Figure 3. First, open the annotation software AnyLabeling NGUYEN et al. (Figure 3 (a)). The left side shows annotation options, including commonly used polygon annotation tools. The center one displays the image to be annotated. The right side shows possible semantic categories and the current annotation directory. Next, annotate objects such as airplanes (Figure 3 (b)). The image is zoomed to an appropriate size. We use the polygon tool, clicking points along the airplane's contour accurately (Figure 3 (c) (d)). Then, annotate the next object, such as the car (Figure 3 (e)). After annotating all objects, the visualization of the annotation results is shown in Figure 3 (f).

To improve annotation efficiency and maintain continuity between frames, the annotations from a single frame can be reused for subsequent frames. In this approach, the labels from the previous frame are applied directly to the next frame, with minor adjustments made for any moving objects. This method ensures temporal consistency across frames while significantly enhancing the speed of annotation.

## 3.2 CHALLENGES

Due to the specificity of semantic objects and imaging environments in airport scenes, airport semantic segmentation faces numerous challenges. Such as extreme multi-scale, intra-class diversity, inter-class similarity, irregular targets, object occlusion, class imbalance, illumination changes, and weather variations. We highlight several key challenges as follows:

**Extreme Multi-scale**: Multi-scale is a common problem in computer vision, but particularly severe in airport scenes. Due to the long imaging distances, objects in airports are imaged across a wide range of sizes. For example, "vehicle", "airplane" and "people" are often imaged as blurred,

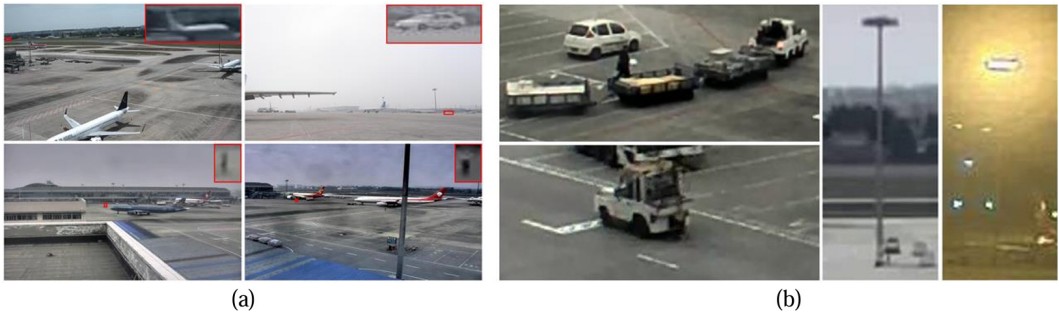

Figure 4: Examples of extreme multi-scale (a) and inter-class diversity (b).

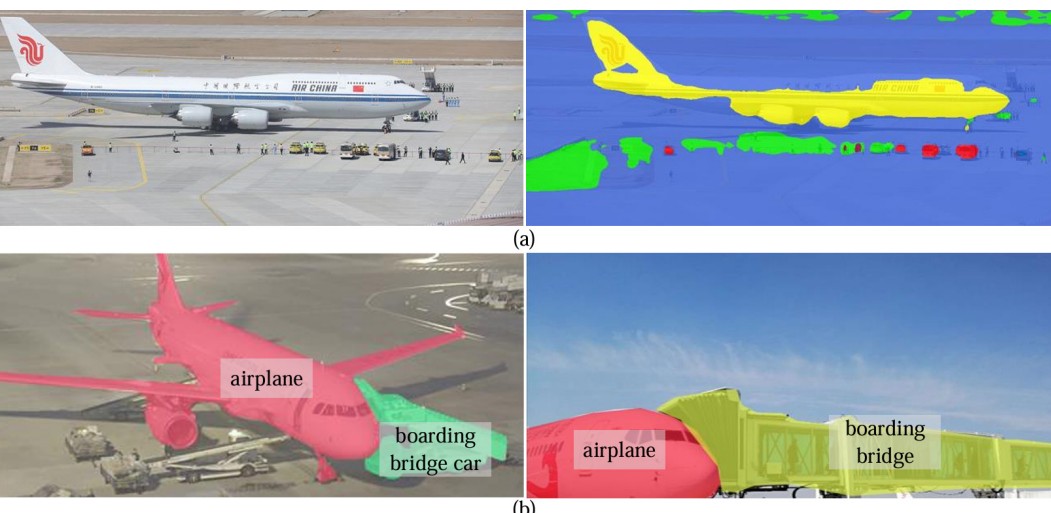

Figure 5: Example of inter-class Similarity. (a) shows the original image and prediction, and (b) represents the boarding bridge and boarding bridge car adjacent to the airplane

small-scale pictures(Figure 4(a)). Comparing "people" in AVSS with pedestrians in other datasets, the semantic property "people" in airports changes. This makes existing methods unsuitable for airport segmentation. Large-scale static objects like "airport terminal" and "sky" dominate the scene. The vast scale difference between semantic categories in airports is termed extreme multi-scale.

**Intra-class Diversity**: The same semantic category can vary significantly in color and shape due to illumination changes, like "airplane" appearing different in daytime and nighttime. And "tractor" vary in shape when towing or not towing loads. The appearance pattern of the "pole" shows a large variation during the day (Figure 4(b)). During daylight without artificial illumination the outlines are distinct and the color appears gray. At night under illumination the halo effect blurs the outlines and shifts the overall color to yellow.

**Inter-class Similarity**: Different semantic categories may have similar appearance and relationships. For example, segmenting gray-white airplanes and aprons is challenging due to similar colors and spatial adjacency(Figure 5 (a)). Likewise, it's different to distinguish between the boarding bridges and the boarding bridge car when them are adjacent to airplanes(Figure 5 (b)). The same problem exists with the segmentation of airplanes and boarding bridges. When spatially close, the likelihood of misclassification increases.

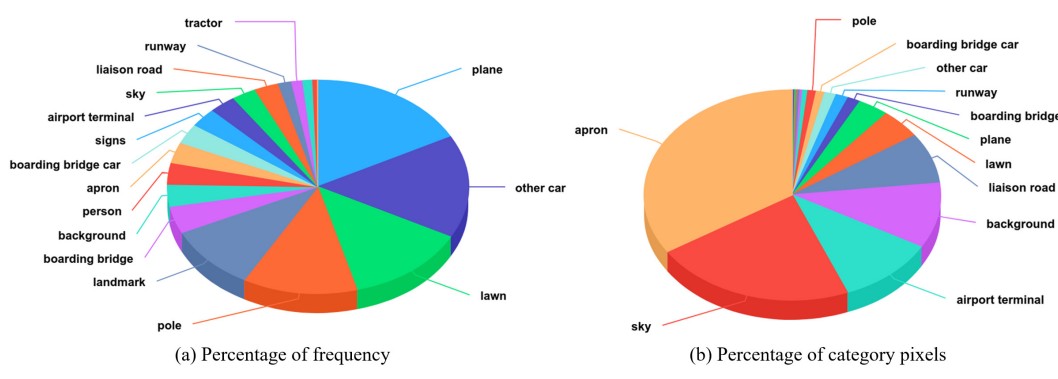

(a) Percentage of frequency          (b) Percentage of category pixels

Figure 6: Distribution of the categories in AVSS. (a) shows the percentage of the instances and (b) shows the percentage of category pixels.

## 4 EXPERIMENTS

In the experimental section of the paper, we explore the value of AVSS from four aspects: statistical analysis, algorithm performance evaluation, generalizability, and future research discussion. First, we assess the diversity and representativeness of AVSS by analyzing statistical information such as sample size, class distribution, and instance compactness. We also check video coherence to ensure the quality. Next, we conduct benchmarking on AVSS, comparing the performance of existing image/video semantic segmentation algorithms and visualizing the segmentation results. Then, we evaluate the performance of models trained on AVSS on other datasets, as well as the performance differences of the same model across various datasets, to highlight the uniqueness and advantages of AVSS. Finally, we discuss the causes of the challenges and propose a preliminary solution.

### 4.1 STATISTICAL ANALYSIS

We analyze the statistical information of AVSS from three aspects: class distribution, video coherence, and compactness.

**class distribution**: We calculate the frequency of each instance category and the proportion of category pixels. The diversity and practicality of AVSS are illustrated in Figure 6 (a), where the variety of instance types and the sufficient number of airport-specific instances are evident. In Figure 6 (b), most pixels in AVSS belong to static objects, while moving objects constitute a smaller portion of the images, such as planes and cars. This indicates the challenges posed by small targets and multi-scale issues with moving objects in AVSS.

**Compactness**: We define a compactness metric to measure the segmentation difficulty of instance categories in the airport, as shown in the following formula, and the results are illustrated in Figure 7.

$$Compactness = \frac{Pixel\ Proportion \times Area}{Minimum\ Bounding\ Circle\ Area}(Pixel\ Proportion = \frac{Class\ Pixel}{Total\ Pixel}),$$

(1)

As compactness decreases, IoU also declines, indicating a positive correlation between compactness and segmentation difficulty.

### 4.2 ALGORITHM PERFORMANCE EVALUATION

**Parameter Settings**: Given the unique semantic categories in airports, existing supervised segmentation models trained on public datasets need to be fine-tuned on AVSS. We set the initial learning rate to 1e-4, the learning rate adjustment strategy to "poly", weight decay coefficient to 5e-4,

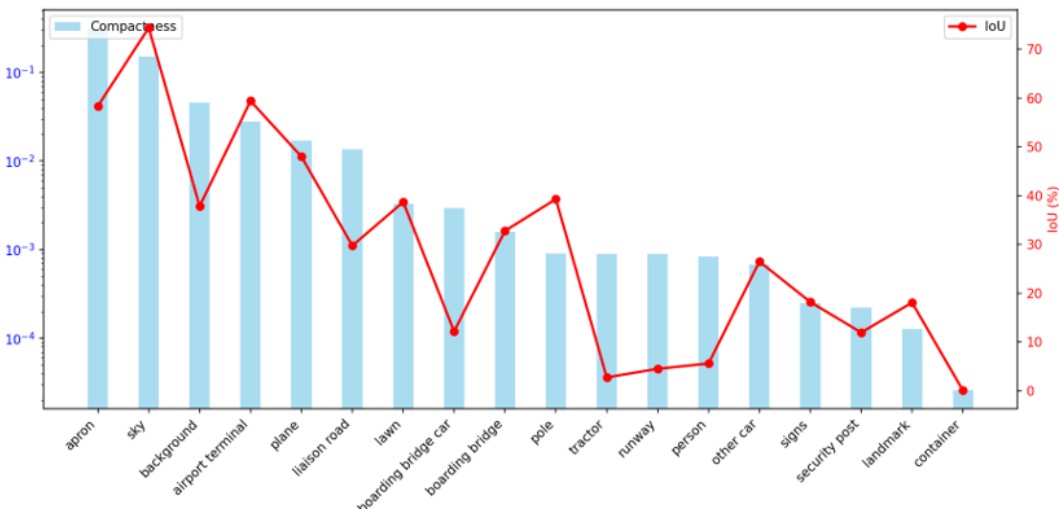

Figure 7: Compactness of instance categories.

fine-tuning epochs to 20k, batch size to 8, and loss function to cross-entropy. Images are resized to 1280×960 for training and testing. Data augmentations include random horizontal flipping, random scaling, random cropping, and color jitter.

Table 1: The 12 classes IoU of 16 SOTA image semantic segmentation models on AVSS.

| Classes
Models | Sky | Liasion Road | Lawn | Apron | Runway | Signs | Landmark | Terminal | Security Post | Airplane | Pole | Person |
|---|---|---|---|---|---|---|---|---|---|---|---|---|
| FCN Long et al. (2015) | 76.40 | 33.61 | 44.91 | 67.89 | 5.36 | 29.76 | 18.27 | 75.06 | 33.65 | 63.34 | 43.70 | 12.82 |
| PSPNet Zhao et al. (2017) | 76.04 | 32.37 | 46.24 | 57.48 | 6.44 | 35.41 | 25.23 | 69.92 | 4.61 | 61.88 | 47.31 | 15.38 |
| DeepLabV3 Chen et al. (2017) | 78.09 | 32.60 | 38.38 | **70.90** | 5.92 | 32.66 | 20.49 | 71.68 | 8.00 | 43.09 | 43.09 | 12.83 |
| DeepLabV3+ Chen et al. (2018) | 78.10 | 32.24 | **50.20** | 55.01 | **8.10** | 22.12 | 24.60 | 70.24 | **46.37** | 58.25 | 49.37 | 5.39 |
| OCRNet Yuan et al. (2020) | 80.52 | 30.92 | 44.73 | 67.68 | 4.76 | 33.61 | 18.47 | 72.17 | 24.00 | 58.03 | 48.74 | 9.34 |
| HRNet Wang et al. (2020) | **83.29** | 34.47 | 43.55 | 55.72 | 6.39 | 34.37 | 26.34 | 73.72 | 14.62 | 57.98 | 50.26 | 10.40 |
| BisenetV1 Yu et al. (2018) | 60.95 | 22.67 | 45.05 | 53.97 | 3.46 | 0.03 | 15.22 | 38.34 | 0.00 | 21.81 | 30.46 | 0.00 |
| BisenetV2 Yu et al. (2021) | 74.69 | 28.39 | 33.70 | 50.45 | 3.83 | 11.81 | 18.03 | 40.56 | 0.00 | 39.19 | 38.25 | 4.05 |
| ICNet Zhao et al. (2018) | 75.73 | 31.00 | 29.03 | 57.84 | 2.23 | 2.52 | 3.07 | 48.14 | 0.00 | 29.48 | 29.20 | 0.00 |
| MobileV3 Howard et al. (2019) | 73.39 | 30.79 | 30.23 | 62.88 | 4.29 | 9.80 | 18.52 | 72.51 | 0.00 | 52.81 | 54.81 | 0.00 |
| SETR Zheng et al. (2021) | 57.96 | 16.17 | 27.80 | 52.87 | 0.10 | 0.00 | 8.89 | 40.63 | 0.00 | 24.09 | 11.99 | 0.00 |
| Segformer Xie et al. (2021) | 73.24 | 33.22 | 14.93 | 40.21 | 4.84 | 0.16 | 14.51 | 50.86 | 0.00 | 37.96 | 29.19 | 0.00 |
| Segmentor Strudel et al. (2021) | 79.09 | 30.28 | 35.95 | 60.66 | 0.32 | 0.00 | 7.74 | 50.11 | 0.00 | 34.77 | 39.64 | 0.00 |
| Swin Liu et al. (2021) | 77.51 | 34.39 | 42.92 | 58.88 | 6.53 | 36.48 | 28.66 | 77.35 | 41.84 | **70.20** | 62.03 | 15.16 |
| SegNeXt Guo et al. (2022) | 79.92 | 34.47 | 42.00 | 64.22 | 4.89 | **36.91** | 27.10 | **77.56** | 13.08 | 65.68 | **63.98** | 15.95 |
| Twins Chu et al. (2021) | 72.20 | **37.67** | 47.57 | 61.67 | 5.29 | 34.81 | **29.19** | 75.90 | 10.20 | 68.36 | 58.00 | **22.86** |
| Average IoU | 74.82 | 30.95 | 38.57 | 58.65 | 4.55 | 20.03 | 19.02 | 62.80 | 12.27 | 49.18 | 43.75 | 7.76 |

**Image Semantic Segmentation Algorithms**: We fine-tune 16 SOTA segmentation models and test them on AVSS. In Table 1, we show 16 baselines' IoU in 12 major airport categories. Categories like "liaison way", "lawn", "runway", "sign", "landmark", "security post", "airplane", "pole" and "people" show average IoUs below 50%. The results show that current semantic segmentation models cannot achieve satisfactory performance on airport-specific semantic categories. We visualized the segmentation results of image semantic segmentation algorithms on AVSS, as shown in the Figure 8. Observing the images, we found that the current model fails to achieve satisfactory segmentation results on the specific semantics of AVSS.

**Video Semantic Segmentation Algorithms**: The number of existing semantic segmentation algorithms is limited, and the source code for some algorithms is unavailable. We tested five algorithms to the best of our ability, and the results are shown in the Table 2. Similar to Guo et al. (2024), five video semantic segmentation models are tested. Compared to image semantic segmentation, video semantic segmentation that leverages temporal information shows better performance in certain categories.

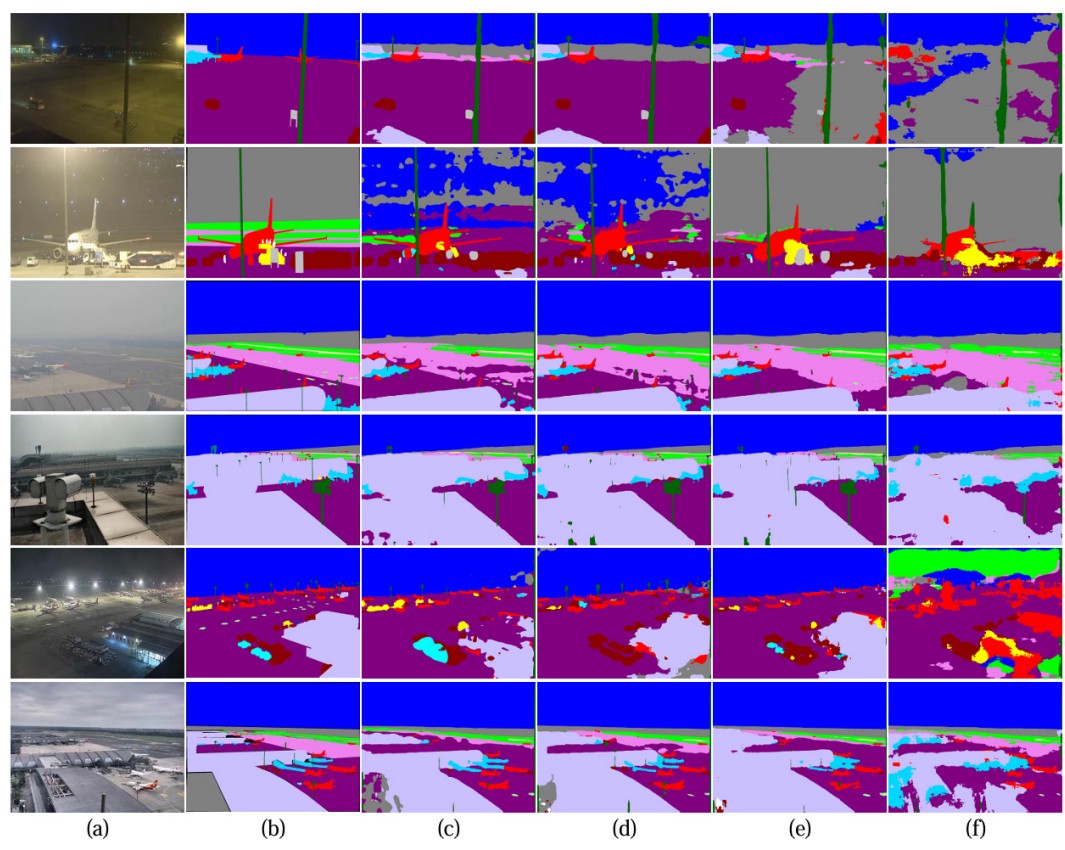

Figure 8: Semantic segmentation visualization results. (a) are original images, (b) are labels, (c), (d), (e), and (f) are the segmentation results of DeepLabV3, FCN, HRNet, and Segformer.

Table 2: The classes IoU of SOTA video semantic segmentation models on AVSS.

| Classes
Models | Sky | Liasion Road | Lawn | Apron | Runway | Signs | Landmark | Terminal | Security Post | Airplane | Pole | Person |
|---|---|---|---|---|---|---|---|---|---|---|---|---|
| **Cfmm** Sun et al. (2022a) | 91.06 | 26.28 | **66.63** | 81.34 | 29.3 | **49.76** | 43.86 | 89.4 | **54.44** | **74.27** | 26.76 | 9.10 |
| **Cfmm++** Sun et al. (2024) | 97.61 | **28.11** | 65.86 | 83.66 | 14.37 | 48.57 | **45.91** | **89.75** | 54.44 | 67.1 | **59.46** | **33.34** |
| **MRCFA** Sun et al. (2022b) | **99.24** | 19.48 | 39.61 | **84.58** | **22.25** | 0.00 | 0.00 | 9.17 | 0.00 | 65.98 | 14.58 | 5.38 |
| **DVIS++** Zhang et al. (2023) | 95.92 | 0.00 | 0.00 | 79.87 | 0.00 | 0.00 | 0.00 | 0.00 | 0.00 | 60.78 | 0.00 | 0.00 |
| **Tube Link** Li et al. (2023) | 82.46 | 27.33 | 38.33 | 69.02 | 39.62 | 0.00 | 0.00 | 37.36 | 0.00 | 33.53 | 6.92 | 29.70 |

## 4.3 GENERALIZABILITY

To demonstrate the generalizability of AVSS, we directly test the model trained on AVSS on VSPW, and the segmentation results of similar categories are presented in Table 3. Similar categories include grass, buildings, sky, vehicles, and pedestrians. For the grass, buildings, sky, and vehicles categories, AVSS-trained models experience a performance decline on VSPW with an average decrease of 27.8%. However, for the pedestrian category, because pedestrians in the airport are considered small and challenging targets, the model's performance on VSPW improved by 28.3% compared to AVSS. These experimental results suggest that the AVSS possesses a certain level of domain adaptability.

Table 3: The classes IoU of SOTA models trained on AVSS, evaluated on AVSS and VSPW.

| Classes
Dataset | Lawn | Building | Sky | Car | Person |
|---|---|---|---|---|---|
| **AVSS** | 37.69 | 30.37 | 74.64 | 29.60 | 6.10 |
| **VSPW** Miao et al. (2021) | 22.78(↓39.6%) | 24.21(↓20.3%) | 62.17(↓16.7%) | 19.32(↓34.7%) | 7.84(↑28.3%) |

To evaluate the difficulty of existing datasets, we compare the performance of 14 baselines on five public datasets and AVSS(Table 4). The public datasets are Cityscapes, ADE20K, Pascal VOC, Pascal Context, and Coco-Stuff 164K. Observing the average mIoU of the 14 models, the worst performance is on AVSS.

Table 4: The mIoU of SOTA models on different datasets. '-' indicates that the current model is not trained on corresponding dataset.

| Models / Datasets | FCN | PSPNet | DeepLabV3 | DeepLabV3+ | OCRNet | HRNet | BisenetV1 | BisenetV2 | SETR | Segformer | Average mIoU |
|---|---|---|---|---|---|---|---|---|---|---|---|
| **Cityscapes** Cordts et al. (2016) | 72.25 | 77.85 | 79.09 | 79.61 | 77.72 | 77.19 | 74.44 | 73.21 | 78.10 | 76.54 | 76.60 |
| **ADE20K** Zhou et al. (2017) | **35.94** | 41.13 | 42.42 | 42.72 | **37.79** | 36.27 | - | - | 48.28 | 37.41 | 40.24 |
| **Pascal VOC** Everingham et al. (2010) | 67.08 | 76.78 | 76.17 | 75.93 | 74.75 | 72.30 | - | - | - | - | 73.84 |
| **Pascal Context** Mottaghi et al. (2014) | 44.43 | 46.60 | 46.55 | 47.30 | - | 45.14 | - | - | - | - | 46.00 |
| **COCO-Stuff 164k** Lin et al. (2014) | - | 38.80 | 39.38 | - | - | - | 25.45 | - | - | - | 34.54 |
| **AVSS** | 37.62 | **37.05** | **37.32** | **37.62** | **38.93** | **20.27** | **26.44** | **15.52** | **22.79** | **20.39** | **29.40** |

## 4.4 FUTURE RESEARCH DISCUSSION

We discuss the causes of each challenge by combining statistical information with segmentation visualization results and introduce airport scene-specific prior knowledge to address challenges, such as Multi-scale, Moving Semantic Objects and Winged Targets. Specifically, we propose a semantic segmentation algorithm based on airplane 3D models.

**Extreme Multi-scale**: Small moving targets and large stationary ones present different challenges. Distant people disappear from the feature map due to their small size, while closer individuals have shape distortions. In Figure 8 (row 5), distant airplanes are segmented roughly, with incomplete wings and tails due to their small pixel count. In contrast, large objects like the sky and terminal (rows 2 and 3) and apron (rows 5 and 6) achieve higher accuracy. This imbalance is due to extreme multi-scale effects. **Intra-class Diversity**: In Figure 8 (row 4), near poles during the day are segmented accurately, but at night (row 6), performance drops, showing the model's difficulty in handling variations within the same category. **Inter-class Similarity**: Airplanes and aprons have high inter-class similarity, causing accuracy shifts. In Table 1, DeepLabV3 reduces aircraft segmentation accuracy but improves apron accuracy compared to PSPNet, while DeepLabV3+ reverses these trends.

The various special challenges present in airports make general semantic segmentation algorithms unsuitable for airport scenes. Therefore, we propose three feasible solutions as follows.

**Categorical Symbiosis**: Multi-scale segmentation is difficult because the scale variation in the scene is unknown and the processing of different scales is incompatible. Observations have shown that the scale of a target can be predicted based on the background structure. For example, the "sky" is usually located at the far end of the airport scene. While the "apron" is usually located at the near end. When an airplane appears in the sky, it is generally a small target. When an airplane appears on the apron, it is generally a large-scale target. Therefore, a priori information on airport structures can be used to solve multi-scale problems.

**Motion Priori**: Several semantic objects in airports have special motion patterns. For example, airplanes have only two directional patterns, straight ahead and turning. Since airplanes are large mass objects, they must follow fixed routes such as runways and liaison roads, or else they will lead to serious safety problems. This inspires us to model the motion and then use it to guide semantic segmentation.

**Shape Priori**: It has been observed that for a wingspan target such as airplanes, strong shape constraints can be imposed by simple manual interaction. For example, the approximate shape of an airplane can be obtained by drawing two cross-crossing lines connecting the head and tail of the airplane and the two flanks. The semantic segmentation algorithm designed based on cross-shape constraints can effectively improve the segmentation accuracy of the wingspan target.

Concretely, as the most important semantic category in airports, we aim for high segmentation accuracy for airplanes. Therefore, we introduce the prior information of 3D airplane models. We first project the 3D models from multiple angles to create a library of 2D projections. During the training of the semantic segmentation network, we also train an object detection head, which identifies airplane regions in the image. Then model selects the most similar 2D projection from the

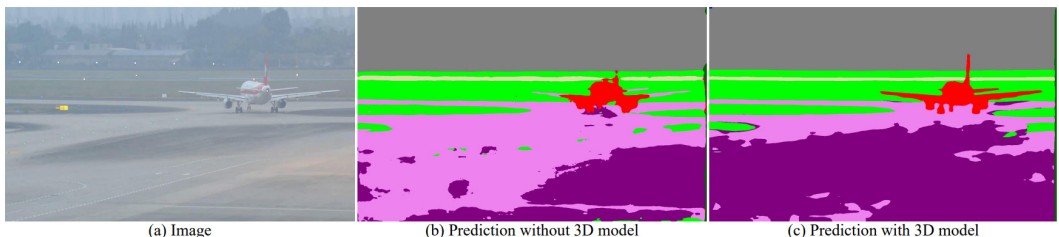

Figure 9: 3D model-based semantic segmentation model for airports.

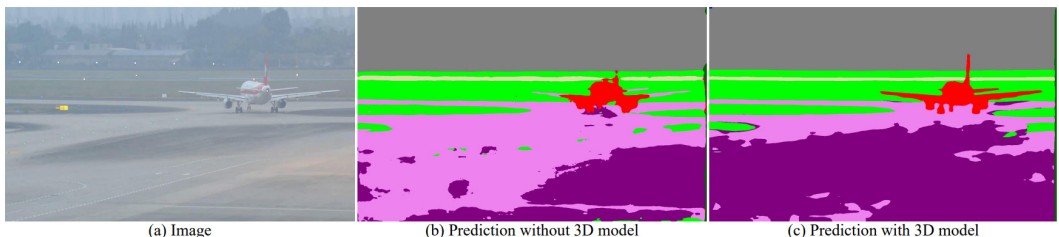

(a) Image        (b) Prediction without 3D model        (c) Prediction with 3D model

Figure 10: Segmentation Performance Comparison.

library to the target area, extracting features from this projection and fusing them with the airplane region features to enhance segmentation accuracy. Figure 9 shows the model schematic, and Figure 10 illustrates the segmentation performance changes before and after introducing the 3D airplane model. The quantitative results are shown in the Table 5.The improvement in airplane segmentation accuracy indicates that the 3D model enhances 2D segmentation performance.

Table 5: Segmentation accuracy change on AVSS by adding 3D model.

| Classes
Models | Sky | Liasion Road | Lawn | Apron | Runway | Terminal | Airplane | Pole | Person |
|---|---|---|---|---|---|---|---|---|---|
| without 3d | 95.55 | 51.05 | 42.78 | 85.62 | 7.74 | 59.00 | 60.83 | 16.85 | 14.85 |
| with 3d | 97.08 | 62.51 | 54.63 | 88.46 | 52.30 | 74.63 | 62.58 | 18.00 | 19.72 |

## 5 CONCLUSION

In this work, we introduced AVSS, a novel benchmark specifically designed for airport video surveillance. With 250 video sequences and over 140,000 pixel-level annotated images, AVSS addresses key challenges in airport scenes by incorporating prior domain knowledge and proposing a semantic segmentation algorithm based on 3D airplane models. Extensive analyses of category distribution, video coherence, and video attributes demonstrated the benchmark's diversity and robustness. By fine-tuning and evaluating 21 state-of-the-art image/video segmentation models, we highlighted the complexity of AVSS. Additionally, comparative studies across different datasets validated the generalizability of AVSS. Overall, AVSS serves as a valuable resource for advancing research in semantic segmentation of airports.

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
