# OpenReview forum: "AVSS: a new benchmark for airport video semantic segmentation"
_ICLR.cc/2025/Conference — Submitted to ICLR 2025_

### Official Review · Reviewer_2Cst · 2024-11-01

**Soundness:** 3
**Presentation:** 3
**Contribution:** 3
**Rating:** 6
**Confidence:** 4

**Summary:**

This paper introduces a new dataset for airport video semantic segmentation (AVSS) with manually labeled masks. It then benchmarks current VSS models on this dataset, highlighting a significant drop in performance when applied to AVSS and providing future insights for designing suitable VSS models for this domain.

**Strengths:**

+ The dataset is based on novel airport scenes, providing a new perspective to evaluate VSS models and contributing a valuable resource for future researchers.
+ The dataset is manually labeled, ensuring high-quality annotations.
+ The experiments reveal the limitations of existing models on this dataset.

**Weaknesses:**

- The scale of AVSS (only 250 videos) is relatively small compared to current VSS datasets.
- The related work section should include more recent advancements in VSS models and benchmarking should add some recent works, such as:
   - Mask propagation for efficient video semantic segmentation
   - Pay attention to target: Relation-aware temporal consistency for domain adaptive video semantic segmentation
   - Temporal-aware Hierarchical Mask Classification for Video Semantic Segmentation

**Questions:**

Refer to weakness.

---

> ### Author Response · Authors · 2024-11-19
>
> For Weakness 1: The scale of AVSS (only 250 videos) is relatively small. Since the airport is a single type of environment, while other VSS datasets are not limited to one scene, the airport videos we captured contain a significant amount of repetitive footage. To ensure data coverage and diversity, the 250 video clips we retained are sufficient to represent the majority of the movements within the airport.
>
> For Weakness 2: The related work section should include more recent advancements. We will cite more VSS models in the related work section of the manuscript, including these three papers. Additionally, we are currently training more VSS models on the AVSS dataset.

---

> > ### Comment · Reviewer_2Cst · 2024-11-25
> >
> > Thank you for addressing my questions. I will keep my current score.

---

### Official Review · Reviewer_UnUY · 2024-11-03

**Soundness:** 3
**Presentation:** 3
**Contribution:** 2
**Rating:** 5
**Confidence:** 3

**Summary:**

In this paper, it introduces the first large-scale Airport Video Semantic Segmentation dataset (AVSS) for airport surveillance. AVSS comprises 18 common semantic categories at airports, and 250 videos, totaling over 140,000 frames with accurate manual annotations. AVSS covers a wide range of challenges for airport video surveillance, such as extreme multi-scale, intra-class diversity, inter-class
similarity, etc. The authors analyze statistical information and evaluate 18 state-of-theart (SOTA) semantic segmentation algorithms on AVSS. The significant performance degradation indicates that current models are far from practical application. Furthermore, this paper discuss how to develop video semantic segmentation algorithms for airport surveillance and the generalizability of AVSS to other tasks
and datasets. AVSS serves as a research resource for airport semantic segmentation and a robustness evaluation tool for segmentation algorithms in practical applications.

**Strengths:**

In conclusion, the main contributions are as follows:
1. this paper establish the novel AVSS, providing a benchmark for airport semantic segmentation.
2. this paper evaluate the generalizability of AVSS and 18 SOTA segmentation algorithms on AVSS.
3. this paper propose principles for designing airport video semantic segmentation models.

**Weaknesses:**

The weakness can be as follows:

1. In table 1, the paper shows the coherence is higher than other datasets. But for the cohenrence metric equaiton (1) (2), it's not clear how to compute the corresponding pixels between difference frames; in addition, higher coherence means the data variety is low for a video squezze, as the AVSS datasets only has 250 videos, does it mean it has only 250 scenes with just changing people, plane etc but the background is fixed. From this view, the variety of the AVSS dataset can be low.
2. For the 4.3 GENERALIZABILITY,  the first sentence is "test the model trained on AVSS on VSPW", is it a typo? As the table 4 shows "The classes IoU of SOTA models trained on AVSS, evaluated on AVSS and VSPW." For this experiment, it is necessary to conduct the experiment comparing the model trained on VSPW and test on AVSS and VSPW.

**Questions:**

No.

---

> ### Author Response · Authors · 2024-11-19
>
> For Weakness 1: For the coherence metric, this metric does not effectively reflect the advantages of our dataset, so we will remove it. Regarding the diversity of the dataset, since the airport is a single type of environment with high video redundancy, we made every effort to collect videos from different scenes across five airports. Ultimately, we retained 250 video clips, carefully considering factors such as weather, lighting, and camera perspectives. In the future, we will collect data from China's first simulated airport to enhance diversity.
>
> For Weakness 2: The comparison of the model trained on VSPW and the test on AVSS and VSPW. For this experiment, it is necessary to conduct the experiment comparing the model trained on VSPW and test on AVSS and VSPW. We chose the CFFM model, trained it on the VSPW dataset, and then evaluated its performance on similar categories in both the AVSS and VSPW datasets. The experimental results are shown in the table below. From the results, it can be observed that, for the five categories, the model's performance on AVSS decreased compared to VSPW. This is because the airport scene is relatively homogeneous. Except for the sky category, where the differences between the two datasets are minimal, other categories show significant disparities. For instance, the building and lawn categories in the airport scene are located in the far distance, with unclear color and shape information, resulting in a sharp decline in segmentation performance. Additionally, categories like car and person dominate in the VSPW dataset, where they are not considered small objects, whereas in the airport setting, these categories are very small. This leads to a large intra-class variation between the two datasets, which ultimately causes worse segmentation results. Therefore, the model achieves relatively good segmentation performance on the lawn, building, and sky categories. However, for the car and person categories, the performance is less satisfactory due to the specific characteristics of the airport scene, where the appearance patterns of the same semantic categories change, leading to suboptimal results on AVSS.
> | Dataset | Lawn  | Building | Sky   | Car   | Person |
> |---------|--------|----------|-------|-------|--------|
> | VSPW    | 77.68  | 46.83    | 95.52 | 56.59 | 82.55  |
> | AVSS    | 20.29  | 10.79    | 83.42 | 0.0   | 0.0    |

---

### Official Review · Reviewer_vXhQ · 2024-11-03

**Soundness:** 2
**Presentation:** 2
**Contribution:** 2
**Rating:** 3
**Confidence:** 2

**Summary:**

This paper proposes a novel benchmark for Airport video semantic segmentation and introduces a semantic segmentation algorithm based on a 3D airplane model. The authors identify key challenges in airport scenes, including extreme multi-scale variation, intra-class diversity, and inter-class similarity. Addressing these issues, they propose a benchmark that is evaluated from various perspectives. First, they conduct a statistical analysis by measuring class distribution, inter-frame coherence, and compactness. By applying the proposed method to various models, the authors demonstrate the increased difficulty of their benchmark compared to existing public datasets. To further assess generalizability, they train models on the proposed dataset and compare performance with the VSPW dataset. To this end, the authors present a 3D airplane model-based algorithm tailored specifically for airport segmentation.

**Strengths:**

1. It provides insights into potential issues in airport semantic segmentation, such as extreme multi-scale, intra-class diversity, and inter-class similarity.
2. The proposed benchmark has been evaluated against various state-of-the-art (SOTA) models.

**Weaknesses:**

1. It would be helpful to specify the diversity principles in “Data collection” in detail.

2. A reference to AnyLabeling in the “Data annotation” should be included.

3. The analysis of whether the proposed benchmark can cover intra-class diversity, a key challenge in airport semantic segmentation, is insufficient. It would be beneficial to examine various aspects, such as color distribution and feature distribution within the same category, to provide a more comprehensive analysis.

4. There is a need to analyze whether the proposed benchmark addresses inter-class similarity.

5. I have doubts about the actual relationship between compactness and segmentation difficulty. For example, in Table 2, while there is a large gap in results between "Runway" and "Liaison Road," the difference in compactness is not significant. Additionally, while there is a small gap in results between "Runway" and "Person," the difference in compactness is large.

6. For the generalizability analysis experiment, comparing similar categories from another dataset (such as Cityscape) would be more helpful.

7. The author suggests that a model performing well on AVSS is likely to achieve favorable segmentation results on other datasets (page 8, lines 430-431). However, it would be useful to have an experiment to verify this claim. For example, by comparing the performance differences between the top 3 models with the highest performance and the bottom 3 models with the lowest performance on AVSS, it could provide valuable insights into whether models that perform well on AVSS have a similar tendency on other datasets.

8. There are no qualitative results for the proposed 3D airplane-based algorithm for airport semantic segmentation.

**Questions:**

Please refer to the weakness part.

---

> ### Author Response · Authors · 2024-11-19
>
> For Weakness 1: The diversity principles in "Data collection". Due to the homogeneity of airport environments, the collected 5,000 video clips contained many repetitive scenes. To maximize scene diversity within the limited airport scenarios, we retained only 250 video clips. After collecting sufficient data, we first extracted features using a neural network for clustering. Redundant video segments within each cluster were then manually removed until a relatively stable set was achieved.
>
> For Weakness 2: A reference to AnyLabeling. We will include a reference to AnyLabeling in the revised manuscript. The reference is as follows: NGUYEN, Viet-Anh, Henry, and StellarKnight. AnyLabeling. v0.3.3, 2 Jul. 2023, https://github.com/vietanhdev/anylabeling.
>
> For Weakness 4: We agree that analyzing whether the proposed benchmark effectively addresses inter-class similarity is crucial. In our AVSS dataset, inter-class similarity does exist, which is an inherent characteristic of the unique airport environment—for instance, the similarity between airplanes and aprons. However, the proposed 3D model-based segmentation algorithm partially mitigates this issue. By integrating the 3D model of airplanes, the algorithm constrains the airplane's shape, reducing the likelihood of misclassifying other objects as airplanes. In future work, we plan to conduct a more comprehensive investigation to evaluate the benchmark’s performance in distinguishing between similar classes.
>
> For Weakness 5: Relationship between compactness and segmentation difficulty. Considering class imbalance, we redefined the compactness metric as $Compactness = \frac{Pixel \ Proportion \times Area}{Minimum \ Bounding \ Circle \ Area}
> (Pixel \ Proportion = \frac{Class \ Pixel}{Total \ Pixel})$ and ranked classes from high to low compactness. The results are shown in the table below:
>
> | **Class**              | Apron | Sky   | Background | Airport Terminal | Plane | Liaison Road | Lawn  | Boarding Bridge Car | Boarding Bridge | Pole  | Tractor | Runway | Person | Other Car | Signs | Security Post | Landmark | Container |
> |-------------------------|-------|-------|------------|-------------------|-------|--------------|-------|---------------------|-----------------|-------|---------|--------|--------|-----------|-------|---------------|----------|-----------|
> | **IoU**                | 58.35 | 74.4  | 37.83      | 59.39            | 47.98 | 29.72        | 38.68 | 12.16              | 32.77           | 39.27 | 2.73    | 4.49   | 5.57   | 26.48     | 18.18 | 11.93         | 18.07    | 0.03      |
> | **Compactness** | 0.32  | 0.15168 | 0.04594   | 0.02812         | 0.01707 | 0.01366     | 0.00329 | 0.00296           | 0.00158         | 0.0009 | 0.0009  | 0.0009 | 0.00084 | 0.00068   | 0.00025 | 0.00022       | 0.00013  | 0.00003   |
>
> As compactness decreases, IoU also declines, indicating a positive correlation between compactness and segmentation difficulty.
>
> For Weakness 6: Comparing similar categories with another dataset. We evaluate the performance of the model trained on AVSS on the VSPW dataset for semantic categories similar to those in AVSS. The IoU for these categories decrease by 10-15%, except for the person category, which see a 2% increase. The results are presented in the table below:
>
> | **Dataset** | **Lawn** | **Building** | **Sky** | **Car** | **Person** |
> |-------------|----------|--------------|---------|---------|------------|
> | **AVSS**    | 37.69    | 30.37        | 74.64   | 29.60   | 6.10       |
> | **VSPW**    | 22.78    | 24.21        | 62.17   | 19.32   | 7.84       |
>
> For Weakness 8: The qualitative results for the proposed 3D airplane-based algorithm for airport semantic segmentation are shown as follows:
> | Class       | Sky   | Liasion Road | Lawn  | Apron  | Runway | Terminal | Airplane | Pole  | Person |
> |-------------|-------|--------------|-------|--------|--------|----------|----------|-------|--------|
> | Without 3D  | 95.55 |    51.05     | 42.78 | 85.62  | 7.74   | 59.00    | 60.83    | 16.85 | 14.85  |
> | With 3D     | 97.08 |    62.51     | 54.63 | 88.46  | 52.30  | 74.63    | 62.58    | 18.00 | 19.72  |

---

> ### Author Response · Authors · 2024-11-20
>
> For Weakness 3: The analysis of intra-class diversity. We employed the Coefficient of Variation (CV) to evaluate intra-class diversity in our dataset, focusing on both color distribution and feature distribution.
>
> ### **Definition of CV**
> The Coefficient of Variation (CV) is a statistical measure of data dispersion relative to its mean, expressed as a percentage. It is calculated as:
>
> $$
> CV = \frac{\sigma}{\mu}
> $$
>
> where \( $\sigma$ \) is the standard deviation, and \( $\mu$ \) is the mean. CV is beneficial for comparing variability across datasets with different means.
>
> ### **Methodology**
> - **Color Distribution:** CV was computed for each RGB channel independently by calculating the standard deviation and mean of pixel values within each class.
> - **Feature Distribution:** Using the final layer of a ResNet-18 model, we extracted 512-dimensional feature vectors for each image. CV values were computed for each dimension across all images in a class, and their mean was used to represent overall feature variability for that class.
>
> ### **Experimental Results**
>
> #### **Color Distribution**
>
> | Class                | Red_CV | Green_CV | Blue_CV |
> |----------------------|--------|----------|---------|
> | Background           | 0.74   | 0.72     | 0.73    |
> | Sky                  | 0.31   | 0.30     | 0.30    |
> | Liaison Road         | 0.33   | 0.31     | 0.32    |
> | Lawn                 | 0.33   | 0.30     | 0.32    |
> | Apron                | 0.31   | 0.31     | 0.33    |
> | Runway               | 0.30   | 0.29     | 0.31    |
> | Signs                | 0.48   | 0.42     | 0.39    |
> | Airport Terminal     | 0.25   | 0.21     | 0.20    |
> | Boarding Bridge      | 0.42   | 0.39     | 0.38    |
> | Other Car            | 0.36   | 0.34     | 0.33    |
> | Tractor              | 0.64   | 0.62     | 0.60    |
> | Boarding Bridge Car  | 0.48   | 0.48     | 0.46    |
> | Security Post        | 0.47   | 0.47     | 0.46    |
> | Pole                 | 0.37   | 0.36     | 0.32    |
> | Plane                | 0.48   | 0.47     | 0.46    |
> | Person               | 0.33   | 0.36     | 0.35    |
> | Container            | 0.35   | 0.34     | 0.31    |
>
> #### **Feature Distribution**
>
> | Class                | Average CV |
> |----------------------|------------|
> | Background           | 0.99       |
> | Sky                  | 1.20       |
> | Liaison Road         | 0.99       |
> | Lawn                 | 1.10       |
> | Apron                | 0.83       |
> | Runway               | 1.46       |
> | Signs                | 1.06       |
> | Airport Terminal     | 1.31       |
> | Boarding Bridge      | 1.45       |
> | Other Car            | 1.63       |
> | Tractor              | 1.11       |
> | Boarding Bridge Car  | 1.36       |
> | Security Post        | 1.36       |
> | Pole                 | 1.09       |
> | Plane                | 1.52       |
> | Person               | 0.93       |
> | Container            | 1.65       |
>
> ### **Insights**
> The analysis highlights significant intra-class variability:
> - **Color Distribution:** Even classes with relatively low variability, such as "Sky," "Runway," and "Airport Terminal," exhibit CV values of approximately 0.30, indicating notable differences.
> - **Feature Distribution:** All classes show CV values exceeding 0.3, with some exceeding 1.5, such as "Other Car" and "Container," indicating substantial feature variability.
>
> These results emphasize each class's inherent diversity and complexity, posing challenges for consistent feature extraction and segmentation in airport scenes.
>
>
> For Weakness 7: For Weakness 7: Regarding this conclusion: A model performing well on AVSS is likely to achieve favorable segmentation results on other datasets (page 8, lines 430-431). In this paper, we will remove this conclusion. However, as shown in Table 2, we have conducted similar experiments comparing the performance of several models across different datasets. From the experimental results, it can be observed that due to the unique characteristics of the airport environment, models that perform poorly on AVSS do not necessarily perform poorly on other datasets. Similarly, models that perform well on AVSS do not demonstrate significant advantages on other datasets.

---

> > ### Comment · Reviewer_vXhQ · 2024-11-22
> >
> > We appreciate your time and effort in addressing the feedback. Your responses have provided deeper understanding of the intentions behind your research. However, certain concerns remain unresolved, particularly the analysis of inter-class diversity (Weakness 4), the relationship between compactness and difficulty (Weakness 5), and the evaluation of model performance on other datasets (Weakness 7). Therefore, we have decided to maintain the current score.
> >
> > Once again, thank you for your efforts, and we wish you continued success in your research endeavors.

---

> > > ### Author Response · Authors · 2024-11-23
> > > **For Weakness 5: Relationship between compactness and segmentation difficulty.**
> > >
> > > We reorganize the presentation of compactness and IoU by sorting compactness in descending order to observe changes in IoU. Theoretically, as compactness decreases, IoU also decreases because more compact objects are generally easier to segment. In the actual results, although IoU does not strictly decline with decreasing compactness, it shows an overall downward trend. This indicates a positive correlation between compactness and segmentation difficulty.
> > >
> > > ![IoU and Compactness per Class](blob:https://github.com/168f964a-ddf3-4b42-bfbb-b26a2620c484)
> > >
> > > | Class               | Compactness | IoU    |
> > > |---------------------|-----------------------|--------|
> > > | apron               | 0.32                 | 58.35  |
> > > | sky                 | 0.15168              | 74.4   |
> > > | background          | 0.04594              | 37.83  |
> > > | airport terminal    | 0.02812              | 59.39  |
> > > | plane               | 0.01707              | 47.98  |
> > > | liaison road        | 0.01366              | 29.72  |
> > > | lawn                | 0.00329              | 38.68  |
> > > | boarding bridge car | 0.00296              | 12.16  |
> > > | boarding bridge     | 0.00158              | 32.77  |
> > > | pole                | 0.0009               | 39.27  |
> > > | tractor             | 0.0009               | 2.73   |
> > > | runway              | 0.0009               | 4.49   |
> > > | person              | 0.00084              | 5.57   |
> > > | other car           | 0.00068              | 26.48  |
> > > | signs               | 0.00025              | 18.18  |
> > > | security post       | 0.00022              | 11.93  |
> > > | landmark            | 0.00013              | 18.07  |
> > > | container           | 3e-05               | 0.03   |

---

> > > ### Author Response · Authors · 2024-11-23
> > > **For Weakness 7**
> > >
> > > In this paper, we will remove this conclusion. However, we conducted relevant experiments to verify the validity of this conclusion, and the experimental results are shown in the table below. The last row of the table represents the performance of different models on the AVSS dataset, while the other rows correspond to the performance of these models on various other datasets. From the table, it is evident that due to the unique characteristics of the airport environment, models that performed poorly on the AVSS dataset, such as SETR and Segformer, do not exhibit poor performance on other datasets. Conversely, models that achieved better results on the AVSS dataset, such as OCRNet and DeepLabV3+, do not demonstrate significant advantages on other datasets. Therefore, the conclusion drawn in the paper may not be entirely accurate, and we have decided to remove this conclusion.
> > > | **Datasets**            | **FCN** | **PSPNet** | **DeepLabV3** | **DeepLabV3+** | **OCRNet** | **HRNet** | **BisenetV1** | **BisenetV2** | **SETR** | **Segformer** | **Average mIoU** |
> > > |--------------------------|---------|------------|---------------|----------------|------------|-----------|---------------|---------------|-----------|---------------|------------------|
> > > | **Cityscapes**          | 72.25   | 77.85      | 79.09         | 79.61          | 77.72      | 77.19     | 74.44         | 73.21         | 78.10     | 76.54         | 76.60            |
> > > | **ADE20K**              | 35.94   | 41.13      | 42.42         | 42.72          | 37.79      | 36.27     | -             | -             | 48.28     | 37.41         | 40.24            |
> > > | **Pascal VOC**          | 67.08   | 76.78      | 76.17         | 75.93          | 74.75      | 72.30     | -             | -             | -         | -             | 73.84            |
> > > | **Pascal Context**      | 44.43   | 46.60      | 46.55         | 47.30          | -          | 45.14     | -             | -             | -         | -             | 46.00            |
> > > | **COCO-Stuff 164k**     | -       | 38.80      | 39.38         | -              | -          | -         | 25.45         | -             | -         | -             | 34.54            |
> > > | **AVSS**                | 37.62   | 37.05      | 37.32         | 37.62          | 38.93      | 20.27     | 26.44         | 15.52         | 22.79     | 20.39         | 29.40        |

---

> > > ### Author Response · Authors · 2024-11-23
> > > **For Weakness 4: Analysis for inter-class similarity.**
> > >
> > > We acknowledge the reviewer's comment on the necessity of analyzing how well our benchmark addresses inter-class similarity.
> > > Indeed, our AVSS dataset inherently exhibits inter-class similarity due to the unique characteristics of the airport environment.
> > > For instance, the visual similarity between airplanes and aprons presents a significant challenge in segmentation tasks.
> > >
> > > In our current work, we have implemented a 3D model-based segmentation algorithm that utilizes a 3D model of airplanes to enhance segmentation accuracy.
> > > This approach specifically aids in accurately distinguishing airplanes from visually similar objects such as aprons by constraining the shape of the airplane in the segmentation process.
> > > Preliminary results demonstrate a significant reduction in misclassification of these objects, highlighting the effectiveness of integrating 3D models in mitigating issues related to inter-class similarity where airplanes are involved.
> > >
> > > However, we recognize that our current method primarily addresses the similarity issues related to airplane features.
> > > Other similar classes, such as boarding bridges and their adjacent vehicles, have not been directly tackled with the same approach in our current framework.
> > > Moving forward, we plan to expand our methodology by incorporating additional 3D models for different objects commonly found in airport environments.
> > > This expansion will allow us to comprehensively address a broader range of inter-class similarities.

---

> > > ### Author Response · Authors · 2024-11-26
> > >
> > > # For Weakness 4: Analysis for inter-class similarity.
> > >
> > > The following content serves as an addendum to our earlier rebuttal,
> > > further elaborating on the methodologies employed and outlining our future plans for enhancing the analysis of inter-class similarity within our AVSS dataset.
> > >
> > > The method for calculating inter-class similarity involves processing masks to identify the edges of various semantic classes and calculating color histograms for these edges in images.
> > > For each class in an image, the method applies binary dilation to the masks, subtracts the original mask to highlight boundary pixels, and computes histograms for RGB channels from these edge pixels in images.
> > > This captures the distribution of color values at the class boundaries across multiple images.
> > >
> > > To assess the similarity between classes, we use Pearson correlation coefficients between their edge color histograms, averaging the results across the RGB channels.
> > > Additionally, a check for physical adjacency between classes is performed using binary dilation; similarity calculations are only made for classes that are adjacent in the images.
> > > The final output is a similarity matrix where each element represents the visual similarity between two classes based on their boundary colors, providing insights into how similar the classes appear in their visual context, as shown in the table below.
> > >
> > > |                | liaison road | lawn | apron | runway | signs | airport terminal | boarding bridge | tractor | boarding bridge car | security post | pole | plane | person | container |
> > > |----------------|--------------|------|-------|--------|-------|------------------|-----------------|---------|---------------------|-------------|------|-------|--------|-----------|
> > > | **liaison road**   | **1.00**      | 0.68 | 0.89  | 0.89   | **0.97** | 0.84    | 0.88   | 0.73    | 0.80     | 0.70    | 0.46 | 0.90  | 0.86   | 0.62      |
> > > | **lawn**           | 0.68         | **1.00** | 0.30  | 0.34   | 0.65  | 0.28    | 0.46    | 0.47    | 0.23     | 0.11     | 0.06 | 0.38  | 0.48   | 0.46      |
> > > | **apron**          | 0.89         | 0.30 | **1.00** | **0.99**  | 0.86  | **0.95**    | **0.87**      | 0.69   | **0.90**    | 0.83     | 0.54 | **0.94** | 0.83  | 0.53    |
> > > | **runway**      | 0.89      | 0.34 | **0.99** | **1.00** | 0.85 | 0.96   | 0.89        | 0.75   | 0.85           | 0.77     | 0.50 | **0.93** | 0.81  | 0.59    |
> > > | **signs**       | **0.97**   | 0.65 | 0.86 | 0.85  | **1.00** | 0.81         | **0.89**     | 0.75   | 0.83           | 0.77     | 0.50 | 0.87 | 0.86  | 0.54    |
> > > | **airport terminal** | 0.84     | 0.28 | **0.95** | 0.96  | 0.81 | **1.00**      | 0.82        | 0.64   | 0.87           | **0.79**  | 0.40 | 0.88 | 0.71  | 0.47    |
> > > | **boarding bridge** | 0.88     | 0.46 | **0.87** | 0.89  | **0.89** | 0.82        | **1.00**     | 0.86   | 0.71           | 0.71     | 0.37 | 0.87 | 0.87  | 0.76    |
> > > | **tractor**      | 0.73     | 0.47 | 0.69 | 0.75  | 0.75 | 0.64         | 0.86        | **1.00** | 0.39           | 0.31     | 0.24 | 0.71 | 0.67  | 0.86    |
> > > | **boarding bridge car** | 0.80   | 0.23 | **0.90** | 0.85  | 0.83 | 0.87         | 0.71        | 0.39   | **1.00**        | **0.95**  | 0.51 | 0.84 | 0.77  | 0.29    |
> > > | **security post**  | 0.70     | 0.11 | 0.83 | 0.77  | 0.77 | **0.79**       | 0.71        | 0.31   | **0.95**        | **1.00**  | 0.52 | 0.77 | 0.72  | 0.24    |
> > > | **pole**        | 0.46     | 0.06 | 0.54 | 0.50  | 0.50 | 0.40         | 0.37        | 0.24   | 0.51      | 0.52     | **1.00** | 0.63 | 0.51  | 0.16    |
> > > | **plane**       | 0.90     | 0.38 | **0.94** | **0.93** | 0.87 | 0.88         | 0.87        | 0.71   | 0.84      | 0.77     | 0.63 | **1.00** | **0.87** | 0.59    |
> > > | **person**       | 0.86     | 0.48 | 0.83 | 0.81  | 0.86 | 0.71         | 0.87        | 0.67   | 0.77      | 0.72     | 0.51 | **0.87** | **1.00** | 0.69    |
> > > | **container**     | 0.62     | 0.46 | 0.53 | 0.59  | 0.54 | 0.47         | 0.76        | 0.86   | 0.29    | 0.24     | 0.16 | 0.59 | 0.69  | **1.00**    |
> > >
> > > A standout observation from the matrix is the high similarity score between "plane" and "apron," with a score of 0.94. This indicates a very close visual resemblance in terms of color at the boundaries of these two classes.
> > > This proximity in color scores presents a significant challenge for segmentation tasks, as the algorithm needs to effectively distinguish between the airplane,
> > > which is an object of interest, and the apron, which is typically a background element. The close similarity can lead to potential misclassifications, where parts of the apron might be incorrectly labeled as part of the plane or vice versa.
> > >
> > > And we plan to integrate prior information such as 3D models, radar point cloud data, and airport structural information
> > > in our subsequent work to mitigate the impact of inter-class similarity.

---

### Official Review · Reviewer_Ed9q · 2024-11-04

**Soundness:** 3
**Presentation:** 3
**Contribution:** 2
**Rating:** 5
**Confidence:** 5

**Summary:**

The paper introduces a new dataset consisting of 250 short videos of outdoors airport scenes, with segmentation annotations of 18 categories on all frames. A variety of recent image and video segmentation methods are fine-tuned and tested on the dataset, yielding relatively low accuracies on most categories. The dataset is characterized by various analytics and a few metrics.

**Strengths:**

Demonstrating the low accuracies of popular segmentation techniques on a real-world problem is a significant contribution. The challenges of this dataset are also found in many surveillance datasets, hence addressing them through research on this dataset could have impact in many surveillance domains.

The distribution of annotations and pixels by class is useful to see, Fig. 5. The dataset seems to follow a long-tail distribution which is very common in real-world settings like this, whereas more contrived datasets often have a relatively uniform distribution.

The annotation compactness metric is well formulated, based on a standard spatial moment calculation. It would be interesting to see a comparison to other surveillance segmentation datasets.

The evaluation is quite thorough, testing a variety of current approaches for both image and video segmentation. The low accuracies of all of the algorithms indicate the difficulty of the problem and the utility of the dataset.

**Weaknesses:**

There is essentially no stated motivation for semantic segmentation at airports. The one mention of this, in the first paragraph is very vague. It would be more convincing to enumerate a set of use cases motivating the need for accurate segmentation rather than bounding boxes.

The Intro should clarify that only outdoor scenes are included in the dataset, since there are numerous video surveillance datasets that include indoor scenes are airports and other large facilities.

Downselecting from 5000 collected videos to 250 included videos is a huge reduction. Was this primarily motivated by the cost of creating ground-truth segmentations? The methods used for data selection are not described.

The data annotation section describes the process for annotating one image, but does not mention how video is annotated. In video from a fixed camera, a single annotation of a fixed object e.g. a Building should be transformable to subsequent video frames without editing. Was this method used? Even for movers, the annotation on the previous frame can be copied to the current frame and adjusted, greatly reducing effort and inter-frame annotation variability.

Creating segmentation annotations manually is expensive on images, even more so on video as performed here. The dataset is much smaller than VSPW in terms of number of videos and classes, partly because of the narrower problem domain.
Image coherence, Eq. 1, does not seem to be an advantage nor disadvantage. With a moving camera, image coherence would be very low, for example. A large number of movers would yield low coherence. Similarly, label coherence is a function of camera and object movement, not just label spatial consistency across frames. The purpose of these measures, and the comparison to other datasets, is very unclear and not well motivated. I’d suggest removing this section, or, more significantly, reformulating these metrics to measure how consistent an object label remains across video frames.

The topic of this paper is rather specific and seems more appropriate for a computer vision venue such as WACV, or AVSS (the conference with the same acronym as the dataset).

**Questions:**

There is a natural hierarchy among some of the classes, such as Building, Terminal (subtype of Building), Tower (subtype of Building). Did you consider defining an explicit class taxonomy rather than just a flat list of classes? A taxonomy would enable natural expansion to additional classes and potentially resolve ambiguities such as incorrectly declaring a false alarm when a Terminal is labeled as a Building.

During data collection, why were the videos so short, 15 sec? Since the cameras are fixed, it seems straightforward to collect long videos e.g. hours in order to capture a diversity of long-range airport activities.

How many airports are in the dataset? How many unique camera views? Many important details are missing.

What proportion of the dataset was used for fine-tuning vs. testing? Were less-frequent classes handled differently than more common ones?

The IOU results on Person are the lowest of any any category, despite the maturity of person detectors. Why? Is it the small pixel size of people in this dataset? What would happen if the images were up-sampled 2X or 4X?

---

> ### Author Response · Authors · 2024-11-18
>
> For Weakness 1: The motivation for semantic segmentation at airports, we believe that semantic segmentation models trained on AVSS can be effectively utilized for tasks such as video docking and collision warning, where precise mask segmentation is critical for preventing accidents. Bounding boxes alone are insufficient for these applications. Moreover, the model can be extended to other scenarios, including crowd management, vehicle monitoring, and similar use cases. By employing semantic segmentation to identify key objects such as vehicles and pedestrians, the system can distinguish between different passenger behaviors, detect abnormal activities to enhance safety, and monitor the movement of various vehicles, ultimately improving both safety and management efficiency.
>
> For Weakness 2: The dataset includes only outdoor scenes. The application scenario of AVSS is focused on airport aprons, with an emphasis on monitoring the movements of aircraft, vehicles, and pedestrians. Consequently, this study exclusively considers outdoor airport environments.
>
> For Weakness 3: A significant reduction from 5,000 collected videos to 250 included videos. Due to the homogeneity of airport environments, the initial dataset of 5,000 video clips contained many repetitive scenes. To ensure diversity, we carefully selected 250 video clips, representing different scenes from five airports. The selection process prioritized factors such as varying weather conditions, lighting, and camera perspectives to maximize the dataset’s representativeness and usability.
>
> For Weakness 4: The video annotation process. We adopted a method that leverages label reuse across adjacent frames to enhance efficiency. Using Anylabeling, labels from one frame can be directly applied to the subsequent frame with a single click, requiring only minor adjustments for moving objects. This approach significantly streamlines the annotation process while maintaining accuracy.
>
> For Weakness 5: The effectiveness of the video coherence metric. Thank you for your question. This metric is influenced by both moving objects and camera motion, making it ineffective at accurately reflecting inter-frame continuity. As a result, we have decided to remove this metric.
>
> For Weakness 6: The relationship between ICLR and database-focused papers. We notice that the list of accepted topics for ICLR explicitly includes "datasets and benchmarks." Therefore, submitting our work to ICLR is both reasonable and aligned with the conference’s scope.
>
> For Question 1: An explicit class taxonomy. We clarify that "Tower" specifically refers to airport control towers, while "Terminal" refers to airport terminals—both are distinct, airport-specific categories. In contrast, "Building" serves as a broader classification encompassing other non-airport-specific structures. During data annotation, "Tower" and "Terminal" are treated as independent categories, enabling the model to recognize the unique architectural features of airports. Meanwhile, "Building" encompasses a wider variety of structures, ensuring the model's generalizability to non-airport contexts. To further enhance precision, airport-specific vehicles such as "Boarding Bridge Car" and "Tractor" are individually annotated, while other vehicles are labeled under the general category of "Car". In the future, we plan to introduce hierarchical constraints to help the model distinguish between specific airport categories and more general types of buildings and objects, improving its ability to adapt to diverse scenarios.
>
> For Question 2: The rationality for short videos. Firstly, long videos significantly increase the manual annotation workload and can reduce annotation accuracy. Secondly, airports often experience extended periods of inactivity, making long videos less relevant for our analysis. Finally, long videos pose challenges in terms of data storage management and computational resources. For these reasons, we opted for shorter videos to maintain efficiency and relevance.
>
> For Question 3: Data sources and camera perspectives. Data were collected from five airports using five fixed cameras and one PTZ camera. In future work, we plan to expand the dataset by collecting additional data from China's first simulated airport.
>
> For Question 4: Dataset splitting and class imbalance handling. Each video segment was randomly sampled and split into an 8:2 training and validation set ratio. No specific measures were implemented to address class imbalance.
>
> For Question 5: Low IoU for person. The low performance for person in AVSS can be attributed to two factors: first, the long shooting distance of the camera, which results in small representations of people; and second, the lack of prominent pedestrian features in airport scenes. Increasing image resolution from 320, 640, 960, to 1280 (equal width and height) improves person segmentation accuracy, with IoU values of 0%, 19.72%, 23.71%, and 34.40%.

---

> > ### Comment · Reviewer_Ed9q · 2024-11-26
> >
> > I thank and applaud the authors for their thorough and energetic rebuttals to all of the reviewers.
> >
> > My main concern is not really rebuttable, since it is inherent in the dataset. The focus on airports is quite narrow, and limits the benefits and community interest in the dataset. Semantic segmentation dataset papers published in premier conferences such as ICLR, CVPR etc. are usually more diverse in scenes, domains, number of classes and so on. This paper would be more appropriate, and find a more receptive and interested audience, in WACV or AVSS.
> >
> > The authors did not address my point about a natural hierarchy in the classes, despite its relationship to class similarity concerns raised by other reviewers. It is generally undesirable to have a generic super-category such as Building, and some of its sub-categories such as Terminal and Tower, in the same (flat) list of categories. In a case like this, Building really means "Buildings that are not Terminals or Towers", which is somewhat unavoidable, and a reasonable compromise. However, I would expect there to be considerable confusion between Building and Terminal which could be measured and taken into account in scoring.
> >
> > The analysis of class similarity, in response to reviewer vXhQ, only considers pixels on the boundaries between classes (if I'm correctly interpreting the description of how it is computed). Consequently this technique does not really measure the color difference between classes, but instead mixes the colors of adjacent classes. Why would this be a useful measure of class similarity? It would be more typical and intuitive to measure class similarity based on feature vectors of class appearance, e.g. exactly the 512-dimensional feature vectors used in the variational analysis.
> >
> > The authors mostly answered my other questions.
> >
> > My score remains unchanged.

---

### Meta-Review · Area_Chair_AT5F · 2024-12-17

**Metareview:**

This paper received ratings of 3, 5, 5, 6 (avg. 4.75).

This paper's core contribution is a video semantic segmentation dataset that is designed for airport surveillance. The dataset consists of 250 videos (totaling ~140k labeled frames) that contain labels for 18 semantic classes. In addition to a statistical analysis of the dataset, the paper evaluates 17 recent models for semantic segmentation, reveals that state-of-the-art models do not perform well on this dataset, and provides suggestions for generalizing prior art to airport scenarios.

Reviewers appreciated the efforts in data collection and labeling and pointed out that demonstrating low accuracy of consolidated methods on a real-world problem is a significant contribution (Ed9q), appreciated through statistical analysis of the dataset, and, overall, note that the dataset provides a new perspective on practical applications of semantic segmentation for future researchers.

However, reviewers also point out that the (specialized) tasks of airport semantic segmentation are poorly motivated and that the dataset lacks diversity (only outdoor scenes of the airport are captured). Reviewers also point out that the reduction of the collected data was not well-justified (as 2Cst
 points out the proposed dataset is relatively small compared to existing video semantic segmentation datasets) and that the paper only explains how individual images were labeled (not the video).  In addition, reviewers also raise concerns about whether ICRL is the right venue for this paper, as it explores a niche topic within Computer Vision.

Ultimately, I agree with Ed9q that the proposed dataset and findings have a too narrow scope (airport surveillance). Three reviewers recommend not accepting this paper (ratings 3, 2x5), and rev. 2Cst (rating of 6) does not present strong arguments for accepting this paper. I agree with the overall reviewer's assessment. A revised version of this paper should also thoroughly address the ethical concerns raised by Rev. vXhQ.

We provide a summary of the review's discussion below.

**Additional Comments On Reviewer Discussion:**

Reviewers and authors engaged in a discussion.

Rev. Ed9q acknowledges that several issues they raised were well addressed (wrt. motivation for airport VSS, data (sub)-selection, outdoor-only coverage). However, despite a good rebuttal, they point out that the core concern was not addressed, as it is inherent in the dataset.

The topic is quite a niche and focuses on studying a (general) problem of video semantic segmentation in a (very) constrained airport environment. It mainly provides scenario-specific, rather than general, insights. The reviewer recommends revising the paper for a computer vision-focused audience and does not increase their rating.

Similarly, VxHq appreciates a detailed author's response and points out several concerns were addressed adequately. However, the reviewer was not convinced with the response to their questions on inter-class diversity and the relationship between the compactness metric and difficulty. Therefore, the reviewer maintains their initial score.

Reviewer UnUY did not respond to author feedback, but they retained their initial ratings.
Reviewer 2Cst (only rating above 5) acknowledged they read the rebuttal and retained their initial rating.

---

### Decision · Program_Chairs · 2025-01-22

Reject